# HiBBO: HiPPO-based Space Consistency for High-dimensional Bayesian Optimisation

## Abstract

Bayesian Optimisation (BO) is a powerful tool for optimising expensive black-box functions, but its effectiveness diminishes in high-dimensional spaces due to sparse data and poor surrogate model scalability. While Variational Autoencoder (VAE)-based approaches address this by learning low-dimensional latent representations, the reconstruction-based objective function often brings the functional distribution mismatch between the latent space and original space, leading to suboptimal optimisation performance. In this paper, we first analyse the reason why reconstruction-only loss may lead to distribution mismatch and then propose HiBBO, a novel BO framework that introduces the space consistency into the latent space construction in VAE using HiPPO—a method for long-term sequence modelling—to reduce the functional distribution mismatch between the latent space and original space. Experiments on high-dimensional benchmark tasks demonstrate that HiBBO outperforms existing VAE-BO methods in convergence speed and solution quality. Our work bridges the gap between high-dimensional sequence representation learning and efficient Bayesian Optimisation, enabling broader applications in neural architecture search, materials science, and beyond.

## 1 Introduction

Bayesian Optimisation (BO) (Brown et al., 2024; Hvarfner et al., 2024) is a powerful sequential strategy for optimising expensive-to-evaluate black-box functions. By leveraging probabilistic surrogate models (typical Gaussian processes (GP) (Williams & Rasmussen, 1995), Bayesian neural networks (Li et al., 2024), or variational Bayesian last layers (VBLLs) (Brunzema et al., 2025)) and acquisition functions to balance exploration and exploitation, BO efficiently guides the search for global optima with minimal function evaluations. This approach is particularly crucial in scenarios where each evaluation is costly or time-consuming, such as hyperparameter tuning in machine learning (Wang et al., 2024), experimental design in materials science (Frazier & Wang, 2015), LLMs in-context optimisation (Agarwal et al., 2025), or controller optimisation in robotics (Yuan et al., 2019). Its data-efficient nature and ability to handle noisy, non-convex objectives have made BO a cornerstone in automated decision-making systems, enabling optimisation in complex, real-world problems where gradient-based methods fail or are infeasible.

While Bayesian optimisation excels in low-dimensional spaces, its performance degrades sharply as dimensionality increases (Ziomek & Ammar, 2023; Moriconi et al., 2020; Lee et al., 2025). This "curse of dimensionality" arises because: (1) the surrogate model's accuracy diminishes exponentially with more dimensions, as data becomes sparse relative to the search space volume; (2) acquisition functions struggle to balance exploration-exploitation in high-dimensional manifolds, often converging to suboptimal solutions; and (3) computational costs of inference and optimisation scale poorly. These limitations hinder applications in modern problems like neural architecture search (Kandasamy et al., 2018) or chemical compound design (Tripp et al., 2020), where data spaces often exceed hundreds or thousands of dimensions.

Variational Autoencoder (VAE) (Kingma & Welling, 2022)-based approaches have emerged as a promising solution to high-dimensional BO by learning compact, low-dimensional latent representations of the input space (Tripp et al., 2020; Ramchandran et al., 2025). These methods first train a VAE to encode high-dimensional data (e.g., molecular structures or neural architectures) into a continuous latent space, where traditional BO is then applied. By exploiting the VAE's ability to capture

Figure 1: The illustration of our idea that is to increase the space consistency between the latent and original spaces by reducing the HiPPO representations of reconstructed data sequence and original data sequence.

meaningful structure and constraints in the data, this approach effectively reduces the optimisation problem's dimensionality while preserving critical features. The quality of BO solutions heavily depends on the VAE's representational capacity, which may discard important information during dimension reduction. To preserve the critical features, a number of strategies were proposed, such as metric learning to enhance the discriminative capability of latent representations (Grosnit et al., 2021), GP prior to model the relationship between data features (Ramchandran et al., 2025), sample reweighting to prioritise the space surrounding the data samples with extreme values (Tripp et al., 2020), and so on. However, one key challenge remains that there is a mismatch between functional distributions (GPs) in the latent space and the original data space. Such a mismatch would make the latent GP not a sufficiently good surrogate model for the following optimisation and the next evaluation points selection.

In this paper, we first analyse the underlying reason for the distribution mismatch that the commonly used reconstruction loss can only preserve the mean but not the kernel distance/relationship between data points. Then, we propose a novel strategy to reduce the distribution mismatch by preserving the kernel distance/relationship during the latent space construction, which is able to further improve the BO performance. As illustrated in Fig. 1, our idea is to use HiPPO (High-order Polynomial Projection Operators) representation (Gu et al., 2020; 2023) to memorise the past (from the beginning to the latest) observations. The restriction on the HiPPO representations from both latent and original spaces could indirectly preserve the kernel distance/relationship between data points. Such a HiPPO representation constraint could enhance the consistency between the constructed latent space with the original data space.

Our contributions can be summarised as follows:

- We introduce the HiPPO-based method to preserve the kernel distance/relationship of data points for the latent space inference from VAE;

- We propose a novel BO algorithm with HiPPO-based space consistency, which is superior to others in a number of standard benchmark tasks.

## 2 BACKGROUND: VAE-BASED BO

Given an unknown objective function $f : \mathbb{X} \to \mathbb{R}$, where $\mathbb{X} \subseteq \mathbb{R}^d$ is the input space, Bayesian optimisation aims to find:

$$x^* = \arg \max_{x \in \mathbb{X}} f(x)$$

where $f(x)$ is computationally expensive to evaluate and has no analytical form. The symbols used throughout the paper and their explanations are listed in a table in Appendix A.1. The process consists of three main steps:

- **Probabilistic Surrogate Model**: A probabilistic model, typically a Gaussian process (GP), approximates $f(x)$. The GP provides a posterior distribution over $f(x)$ based on observed data $D_{1:t} = \{(x_i, y_i)\}_{i=1:t}$, where $y_i = f(x_i) + \epsilon_i$ and $\epsilon_i$ represents noise.
- **Acquisition Function**: An acquisition function $\alpha(x; D_{1:t})$ guides the selection of the next query point $x_{t+1}$. Popular options include Expected Improvement (EI), Probability of Improvement (PI), and Upper Confidence Bound (UCB). The next point is selected as:

$$x_{t+1} = \arg\max_{x \in \mathbb{X}} \alpha(x; D_{1:t}).$$

- **Iterative Process**: The algorithm continuously updates the surrogate model with new observations $(x_{t+1}, y_{t+1})$ as it searches for the global optimum.

As $d$ increases, the volume of the search space $\mathbb{X}$ grows exponentially. This leads to the following issues: the number of points required to cover $\mathbb{X}$ grows exponentially with $d$; The average distance between points in $\mathbb{X}$ increases, making it difficult for the surrogate model to interpolate or extrapolate accurately; Computational complexity is $O(t^3)$, where $t$ needs to be large for high-dimensional space; The kernel function becomes less discriminative, i.e., $\lim_{d\to\infty} \text{Var}[k(x, x')] = 0$; The landscape of the acquisition function $\alpha$ becomes highly multimodal and complex in high dimensions.

The VAE-based dimensionality reduction approach, $\mathbb{Z} \subseteq \mathbb{R}^{d'}$ and $d' \ll d$, for high-dimensional BO can be summarized as: Train a VAE based on current observations, i.e., $\{x_i\}$, to learn mappings: $z \sim \mu_\phi(x)$, $x = \mu_\theta(z)$; Reformulate the optimization problem in the latent space, $z^* = \arg\max_{z \in \mathbb{Z}} g(z)$; Perform BO in the latent space using a GP surrogate model $g(z) \sim GP(m_z, k_z(z, z'))$ and acquisition function $\arg\max_{z \in \mathbb{Z}} \alpha(z|g)$; Map the optimal latent point back to the input space, $x^* = \mu_\theta(z^*)$; Add new observation $(x^*, f(x^*))$ to the data, and repeat.

# 3 OUR METHOD

In this section, we first identify the potential functional distribution mismatch between the reconstructed and original spaces in Section 3.1. Then, in Section 3.2, we propose a HiPPO-based solution to mitigate this mismatch. Finally, in Section 3.3, we introduce a new Bayesian optimisation (BO) algorithm that incorporates this approach.

## 3.1 DISTRIBUTION MISMATCH BETWEEN RECONSTRUCTED AND ORIGINAL SPACE

The inferred GP defined over the latent space could induce a distribution of black-box functions in the original space through the decoder. However, there is a distribution mismatch between reconstructed and original functions, which presents a fundamental challenge when applying VAEs to Bayesian optimisation, because it determines the extent the latent space preserves the information of the original space and also the extent we 'trust' the found optimal value in the latent space.

The standard VAE loss function combines reconstruction error with latent space regularisation:

$$\mathcal{L}(\theta, \phi; x) = \mathbb{E}_{z \sim \mu_\phi(x)} [\|x - \mu_\theta(z)\|_2] + \mathcal{KL}[\mu_\phi(z|x)\|p_0(z)] \tag{1}$$

where $p_0(z)$ is a predefined prior. While this formulation ensures approximate reconstruction and smooth latent representations, it does not guarantee preservation of the relationships between $\{x\}$. When we fit a Gaussian process $g(z) \sim GP(m_z, k_z)$ in the latent space using observations $\{(z_i, f(x_i))\}$, it implicitly defines a GP for the function in the reconstructed space through the decoder mapping:

$$f^{\text{VAE}}(x) = g \circ \mu_\theta^{-1}(x) \sim GP(m_z(\mu_\theta^{-1}(x)), k_z(\mu_\theta^{-1}(x), \mu_\theta^{-1}(x'))).$$

In an ideal scenario where $\mu_\theta$ is perfectly invertible and deterministic, this would exactly match the GP $f(x) \sim GP(m_x(x), k_x(x, x'))$ fitted in the original space, provided the mean and kernel functions satisfy:

$$m_z \circ \mu^{-1}(x) = m_x(x), \quad k_z(\mu_\theta^{-1}(x), \mu_\theta^{-1}(x')) = k_x(x, x').$$

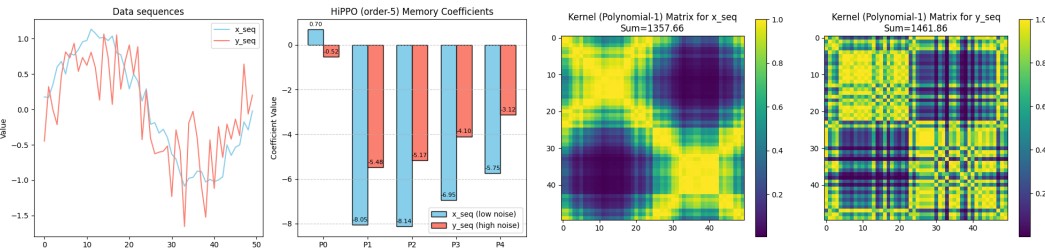

(a) The left subfigure shows two sequences of data points (i.e., *x_seq* and *y_seq*) that have roughly similar correlations due to similar functional trends (the kernel matrix between data points is also visualised in the third and fourth subfigures). As shown in the second subfigure, their evaluated HiPPO representations are close.

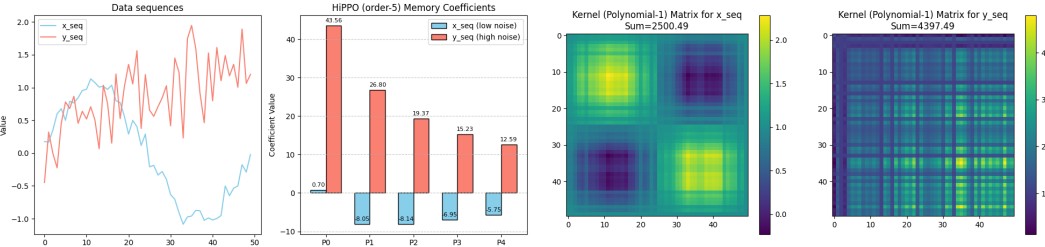

(b) The left subfigure shows two sequences of data points (i.e., *x_seq* and *y_seq*) that have dissimilar correlations due to different functional trends (the kernel matrix between data points is also visualised in the third and fourth subfigures). As shown in the second subfigure, their evaluated HiPPO representations are far from each other.

Figure 2: Empirical demonstration of the capability of HiPPO representation in expressing the correlation between data points.

In practice, the decoder $\mu_\theta$ is neither perfectly invertible nor free from reconstruction errors, leading to inevitable discrepancies between the implied GP of $f^{\text{VAE}}$ and the true GP of $f$. These discrepancies manifest in both the mean and kernel functions:

$$\Delta^{\text{mean}} = \|m_z \circ \mu_\theta^{-1}(x) - m_x(x)\|, \quad \Delta^{\text{kernel}} = \|k_z(\mu_\theta^{-1}(x), \mu_\theta^{-1}(x')) - k_x(x, x')\|.$$

The general reconstruction loss in VAE (i.e., first item of Eq. (1)) could reduce $\Delta^{\text{mean}}$ well but not $\Delta^{\text{kernel}}$. So, even when reconstruction error is minimised, the geometric relationships between points may not be preserved in the latent space, potentially leading the surrogate model to either over-smooth or over-estimate correlations of data. This can cause the acquisition function to either miss promising regions or exploit false optima. It has been unfortunately ignored by the existing literature.

A key challenge in addressing this mismatch is that while we can minimise reconstruction error during VAE training, we cannot directly optimise the kernel discrepancy since the true kernel $k_x$ is unknown - fitting a GP in the original high-dimensional space $\mathcal{X}$ is typically infeasible. This motivates our following approach to indirectly reduce this kernel discrepancy, which helps preserve the critical relationships between data points during the latent space construction. By maintaining better consistency between the original and latent space representations, we aim to reduce both $\Delta^{\text{mean}}$ and $\Delta^{\text{kernel}}$, leading to more reliable optimisation performance.

## 3.2 REDUCE KERNEL DISTANCE VIA HiPPO MEMORY REPRESENTATION

Our idea is to preserve the 'kernel distance/correlation' indirectly with the help of the HiPPO (Gu et al., 2020; 2023), which is a framework for online memory representation in continuous and discrete-time sequences. The key idea behind HiPPO is to project an input signal onto a basis of orthogonal polynomials (e.g., Legendre, Chebyshev) in a way that optimally summarises the history of the signal in a finite-dimensional state.

Given (sequentially obtained) data $\{x(t)\}$, HiPPO wants to maintain a memory state $c_t$ that summarizes its history up to time $t$ by projecting $\{x(t)\}$ onto a basis of orthogonal polynomials $\{\mathcal{P}_n\}_{n=1}^{\rho-1}$

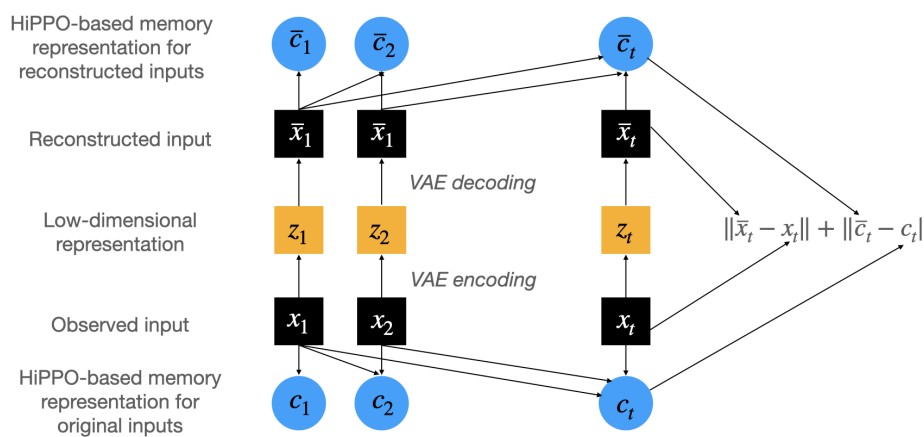

Figure 3: Visualisation of our method.

(e.g., Legendre, Chebyshev, Laguerre) under measure $\omega$, i.e., $\int_{-\infty}^{t} \mathcal{P}_n(\tau)\mathcal{P}_{n'}(\tau)\omega(\tau) = \delta_{nn'}$, where $\rho$ is the order that is the highest degree of the orthogonal polynomials used to approximate the input. Given this projection, the original data can be approximated as:

$$x(\tau) \approx \sum_{n=0}^{\rho-1} c_{n,t}\mathcal{P}_n(\tau)$$

where $\tau \leq t$ and $c_{n,t}$ are the time-varying coefficients defined as

$$c_{n,t} = \int_{-\infty}^{t} x(\tau)\mathcal{P}_{n,t}(\tau)\omega_t(\tau)\mathrm{d}\tau.$$

At any time $t$, $c_t = [c_{0,t}, c_{1,t}, \ldots, c_{\rho-1,t}]$ serves as the HiPPO memory of $\{x(\tau)\}_{\tau=1:t}$, which encodes the functional form of $\{x(t)\}$ (i.e., the latent correlations between $x$). Similarly, we can also build the corresponding HiPPO memory for the reconstructed observations $\{\bar{x}(t)\}$ from VAE mapping: $\bar{c}_t$, which encodes the geometric shape of $\{\bar{x}(t)\}$ from the latent space. Then, we propose the following regularizer for the VAE:

$$\|c_t - \bar{c}_t\|. \tag{2}$$

**Proposition 1.** *If the HiPPO order is larger than the degree of the kernel function, then the closeness of HiPPO representations of two sequences of data would imply their closeness of kernel distance.*

*Proof.* Please see the Appendix for details. $\square$

The above proposition implies that reducing the distance between $c_t$ and $\bar{c}_t$ could help preserve the relationship (i.e., kernel distances) between $\{x\}$ and then reduce the distribution mismatch identified in Section 3.1 (also visualised in Figs. 2a and 2b, and the setting details can be found in Appendix A.3). As we observe more (expensive) function values of the new data points in BO (i.e., the expansion of $\{x(t)\}$), we of course need to update the HiPPO memory to incorporate the new observations. Fortunately, such an update is quite efficient in HiPPO because it is proven that the history $c(t)$ follows a linear ordinary differential equation (ODE):

$$\frac{\mathrm{d}c_t}{\mathrm{d}t} = A_t c_t + B_t x(t) \tag{3}$$

where $A$ and $B$ depend on used polynomials $\{\mathcal{P}_n\}_{n=1}^{d-1}$. Such ODE formulation allows efficient online updates, making it useful for continuing observations from BO.

### 3.3 BO WITH HiPPO-BASED SPACE CONSISTENCY

To reduce the distribution mismatch between the latent space and data space, we propose to use the following new loss function for VAE training:

$$\mathcal{L}(\theta, \phi; x) = \mathbb{E}_{q_\phi(z|x)}\left[\|x - \mu_\theta(z)\|_2 + \|c - \bar{c}\|\right] + \mathcal{KL}[\mu_\phi(z|x)\|p_0(z)] \tag{4}$$

---

**Algorithm 1** HiBBO

---

**Input:** Budget $\mathcal{B}$, frequency $\nu$, some data points $X$
**Output:** optimum value $x^*$
**for** $j \in [1, \lceil \mathcal{B}/\nu \rceil]$ **do**
    *// optimise VAE*
    initialise $c = c_t = 0$ and $\bar{c} = \bar{c}_t = 0$;
    **for** number of epochs **do**
        **for** each $x_i \in X$ **do**
            Reconstruct $\bar{x}_i = \mu_\theta \circ \mu_\phi(x_i)$ ;
            Compute HiPPO memory $c_{t+1}$ (with $x_i$ and $c_t$) and $\bar{c}_{t+1}$ (with $\bar{x}_i$ and $\bar{c}_t$) by Eq. (3);
            $c = [c_t, c_{t+1}], \bar{c} = [\bar{c}_t, \bar{c}_{t+1}]$;
            $c_t = c_{t+1}, \bar{c}_t = \bar{c}_{t+1}$;
        **end for**
        Optimise $\mu_\theta \circ \mu_\phi$ by Eq. (4);
    **end for**
    *// BO in latent space*
    **for** $k = 0$ **to** $\nu - 1$ **and** $\alpha(\hat{z}_{j,k+1}) \geq \eta$ **do**
        Fit surrogate GP on $\langle z_i, f(x_i) \rangle$
        Optimise $\alpha$ for $\hat{z}_{j,k+1}$;
        Use decoder to map $\hat{z}_{j,k+1}$ to $\hat{x}$;
        Evaluate $f(\hat{x})$ to augment data $X = [X, \hat{x}]$;
    **end for**
**end for**
$x^* = \arg\max_{x \in X} f(x)$.

---

where the first item at the right-hand side is now composed of two parts: one is reconstruction loss, and the other is HiPPO-based space consistency constraint (here, we select HiPPO-LegS to keep the whole pass observations in memory and leave the study on the different polynomials to future work). The procedure is visualised in Fig. 3, and the pseudocode in Algorithm 1 summarises our approach. The budget $\mathcal{B}$ refers to the maximum number of evaluations of the unknown functions, and frequency $\nu$ denotes the number of BO steps performed before updating VAE model. There are mainly two procedures in the Algorithm 1. The first procedure is the update of VAE under the HiPPO-based space consistency constraint. We will firstly reconstruct the data $X$ based on VAE, then construct the HiPPO-based memory representations $c$, and finally, the loss function in Eq. (4) is used to guide the optimisation of the encoder and decoder of VAE. Compared with classical VAE training, this new solution is expected to reduce the distribution mismatch between the latent space and data space. The second procedure is the BO steps in the latent space, which is common to all VAE-based BO.

## 4 RELATED WORKS

The literature on VAE-based high-dimensional Bayesian Optimisation (BO) explores methods to efficiently optimise expensive black-box functions in high-dimensional spaces by leveraging VAEs for dimensionality reduction. Traditional BO struggles in high dimensions due to the curse of dimensionality, but VAEs provide a solution by learning a compressed, structured latent space that retains the essential features of the input data. One of the earliest and most influential applications of VAE-based BO was in automatic chemical design, where Gómez-Bombarelli et al. (2018) used a VAE to encode molecular structures into a continuous latent space, enabling gradient-based optimisation of molecular properties. However, this method often generated invalid molecules due to BO exploring regions of latent space far from the training data. To address this, Tripp et al. (2020) proposed constrained Bayesian optimisation, which restricts the search to regions likely to decode into valid molecules, significantly improving the validity and utility of generated samples. Building on this, Grosnit et al. (2021) introduced a method that combines VAEs with deep metric learning to structure the latent space using label information from the black-box function. This approach enhances the smoothness and informativeness of the latent space, improving the performance of the GP surrogate model used in BO. Their method achieved state-of-the-art results on benchmarks

like penalised logP for molecule generation, requiring significantly less labelled data than previous methods.

Recent studies have also explored structured priors, such as GP (Ramchandran et al., 2025)), for the latent variable of VAE to capture the relationship between data and their external features. Meanwhile, Notin et al. (2021) proposed to leverage the epistemic uncertainty of the decoder to guide the optimisation process to improve its robustness and sample validity. Applications of VAE-based BO extend beyond chemistry. In protein design, Zeng et al. (2024) utilised VAE-BO to efficiently generate diverse, high-affinity antibody candidates, which were subsequently validated through in vitro synthesis. Similarly, in materials science, Tian et al. (2024) demonstrated the use of VAE-BO for efficient design of electromagnetic metamaterials by reducing the high-dimensional microstructure space into a compact latent space. VAE-BO was also used in robotics (Antonova et al., 2020) to enable ultra data-efficient controller tuning by learning low-dimensional latent representations of simulated trajectories, eliminating the need for expert-designed features and reducing exploration in undesirable state spaces.

In summary, VAE-based BO is a powerful framework for optimising complex, high-dimensional functions. However, its success depends critically on the quality of the latent space for which this work offers a new strategy.

## 5 EXPERIMENTS

We evaluate the efficacy of our proposed method, **HiBBO**, across four high-dimensional optimisation problems: a standard Bayesian optimisation benchmark function, an MNIST-based synthetic problem, a 2D shape optimisation problem, and a chemical design application. These problems are selected to demonstrate the versatility of HiBBO in handling both synthetic and real-world scenarios while operating effectively in high-dimensional spaces.

For comparison, we consider several VAE-based Bayesian optimisation methods as baselines. **BASE** serves as the simplest baseline, employing a standard VAE without any additional constraints on the latent space. **METRIC** (Grosnit et al., 2021) improves upon this by integrating a triplet loss to enhance the discriminative power of the learned latent representations. In contrast, **GPior** (Ramchandran et al., 2025) replaces the conventional Gaussian prior with a Gaussian Process prior during VAE training, allowing the model to incorporate additional information directly into the latent structure. Finally, **REWEIGH** (Gómez-Bombarelli et al., 2018) modifies the VAE's training objective by reweighting data points to prioritise those with higher performance values.

To ensure a fair and unbiased comparison, all methods share the same VAE architecture for each problem, and training procedures—including optimisation steps, retraining schedules, BO budget, and hyperparameter settings—are kept consistent across experiments. This controlled setup eliminates architectural and training-related confounders, ensuring that performance differences can be attributed solely to the distinct algorithmic contributions of each method.

### 5.1 STANDARD FUNCTION OPTIMISATION

We begin our experimental evaluation with a high-dimensional optimisation benchmark using the Ackley function, implemented through the **BoTorch** framework[1]. The Ackley function represents a challenging multimodal optimisation problem in $d$-dimensional space, typically evaluated on the hypercube $[-32.768, 32.768]^d$. For our experiments, we set $d = 1000$ to create an especially demanding high-dimensional optimisation scenario. This function features numerous local minima but has a single global minimum of 0 at the origin, providing a clear optimisation target for evaluating algorithm performance. The dimension of latent space from VAE is set 10, representing a 100-fold dimensionality reduction from the original 1000-dimensional space. More details of the setup can be found in the Appendix. As shown in Figure 4a, HiBBO demonstrates superior performance compared to baseline methods. Notably, it achieves solutions closest to the global minimum, while also converging faster in terms of iteration count. Since this problem is relatively easy, the differences among various methods are small.

---

[1]https://botorch.org/

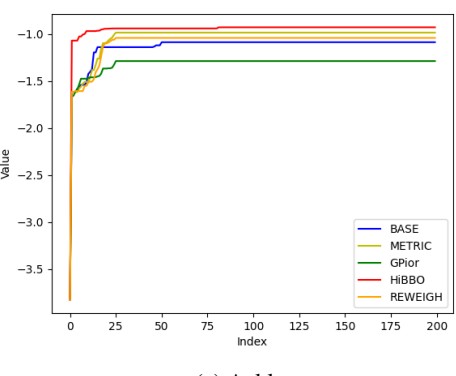
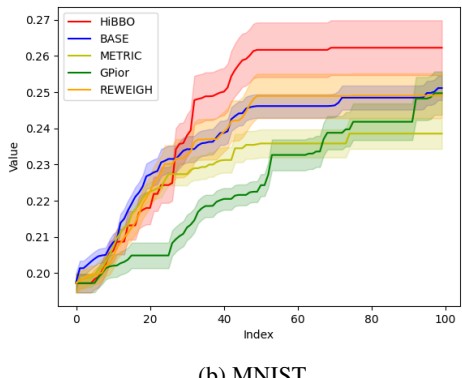

(a) Ackley                              (b) MNIST

Figure 4: results on standard function optimisation and MNIST-based synthetic problem

## 5.2 MNIST-BASED SYNTHETIC PROBLEM

The second benchmark problem adopts the official high-dimensional optimisation example from BoTorch, where the black-box objective function maps MNIST digit images to scalar scores through a carefully designed evaluation pipeline. In this task, the original 784-dimensional search space ($28 \times 28$ pixels) represents handwritten digits from the MNIST dataset, with the optimisation target being the discovery of images minimising a score function that quantifies deviation from the ideal digit '3'. The scoring mechanism first processes each image through a fixed pretrained CNN classifier to obtain a 10-dimensional probability vector $prob$ across digit classes, then computes the weighted inner product $prob \cdot score$ where $score = \exp(-2 \times (v - 3)^2)$ and $v = [0, 1, ..., 9]$. This formulation creates a smooth optimisation landscape with the global minimum (score = 1.0) achieved when the classifier confidently predicts the digit as '3'. As evidenced in Figure 4b (4 seeds), HiBBO outperforms competing methods by consistently generating images with both lower final scores, demonstrating its effectiveness in navigating complex, perception-driven search spaces. The complete experimental setup, including CNN architecture details and optimisation hyperparameters, is documented in the Appendix.

## 5.3 SHAPE OPTIMISATION

Another illustrative task, originally proposed by (Tripp et al., 2020), involves optimising for the shape with the largest total area within the space of $64 \times 64$ binary images—that is, maximising the number of pixels with a value of 1. The underlying generative model is a variational autoencoder (VAE) with a simple convolutional architecture (details provided in the Appendix), and its latent space is two-dimensional, i.e., $\mathbb{R}^2$. Given the low-dimensional nature of the latent space, we perform direct optimisation by enumerating a uniform grid over the range $[-3, 3]^2$ instead of GP, following (Tripp et al., 2020). The results are presented in Fig. 5a (5 seeds). As shown, our proposed method, HiBBO, achieves the best performance compared to baseline approaches. When combined with the reweighting strategy (**HiPPO-RW**), the method still attains a strong final objective value, though slightly lower than that of HiBBO alone.

## 5.4 CHEMICAL DESIGN

This experiment builds upon the molecular optimisation task introduced by Tripp et al. (2020) and further explored by Ramchandran et al. (2025) and Grosnit et al. (2021). We adopt the standardised benchmark from Gómez-Bombarelli et al. (2018), which involves synthesising molecules with the highest penalised water-octanol partition coefficient (logP), using the ZINC250k dataset (Irwin et al., 2012) as the starting point. To ensure valid chemical structures, we use a junction tree VAE (Jin et al., 2018) to generate the latent space. Optimisation is performed using the expected improvement acquisition function. For evaluating $c$ and $c_-$, we use vector representations concatenated from *tree_vecs* and *mol_vecs* obtained from the junction tree VAE encoder, rather than the original

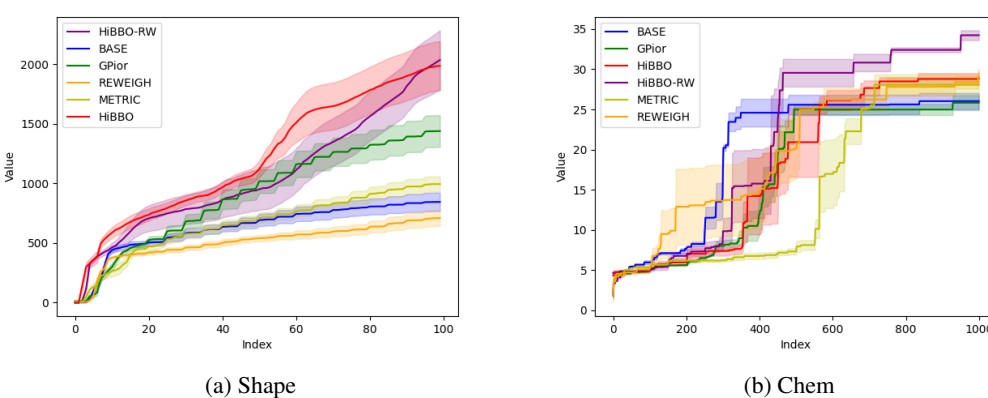

(a) Shape           (b) Chem

Figure 5: results on shape optimisation and chemical design

SMILES strings. All models are trained and evaluated under consistent settings, following the original codebase[2]. Additional implementation details are provided in the Appendix. Fig. 5b (5 seeds) shows the optimisation results. Our method, HiBBO, achieves excellent final objective values, and the HiBBO-RW variant stands out, attaining the highest logP score with the lowest query budget among all compared methods. This reinforces the benefit of incorporating reweighting, particularly in complex design tasks like molecular optimisation.

## 6 CONCLUSION

In this work, we introduced HiBBO, a novel Bayesian Optimisation (BO) framework designed to address the persistent challenge of functional distribution mismatch in VAE-based high-dimensional optimisation. By incorporating HiPPO-based memory representations into the latent space construction, HiBBO preserves not only the mean but also the kernel relationships between data points, which is often overlooked in prior work. Our theoretical analysis and empirical results demonstrate that this added space consistency significantly enhances the fidelity of the surrogate model in the latent space, leading to more reliable and efficient optimisation. The proposed HiPPO-based regularisation is lightweight, generalisable, and compatible with existing VAE architectures, making it a practical enhancement for a wide range of BO applications.

Extensive experiments across diverse domains—including synthetic benchmarks, image-based optimisation, shape design, and molecular property optimisation—validate the effectiveness of HiBBO. Compared to state-of-the-art VAE-BO methods, HiBBO consistently achieves superior performance in terms of convergence speed, final objective value, and query efficiency. Notably, the integration of a data reweighting strategy further amplifies its performance. These results underscore the importance of preserving functional geometry during latent space learning and open new avenues for applying BO in complex, high-dimensional tasks such as neural architecture search, materials discovery, and drug design. Future work may explore alternative polynomial bases in HiPPO, adaptive memory mechanisms, and broader integration with other generative models to further enhance the robustness and scalability of high-dimensional BO.

## REPRODUCIBILITY AND LLMs STATEMENT

The implementation and experimental details are provided in the Appendix, and the code is included in the supplementary materials. Large Language Models (LLMs) were used solely for manuscript polishing and proofreading.

---

[2]https://github.com/cambridge-mlg/weighted-retraining

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

## A APPENDIX

### A.1 NOTATION TABLE

Table 1 is the notation table to demonstrate the notation used in this paper.

### A.2 PROOF OF PROPOSITION 1

To investigate whether the closeness in HiPPO memory representations implies the closeness in average pair kernel distances, we firstly formalise them as follows: given two sequences $\{x_i\}_{i=1}^N$ and $\{\bar{x}_i\}_{i=1}^N$,

- The HiPPO framework projects it onto a basis of orthogonal polynomials (e.g., Legendre) up to some order $\rho$, producing finite-dimensional states $c_x \in \mathbb{R}^\rho$ and $c_{\bar{x}} \in \mathbb{R}^\rho$ that summarise two sequences. Their closeness can be measured by $\|c_x - c_{\bar{x}}\|_2$.

- For a kernel $k(\cdot, \cdot)$ (e.g., linear, Gaussian), the average pair kernel distance for $\{x_i\}$ is $D_x = \frac{1}{N^2} \sum_{m=1}^N \sum_{n=1}^N k(x_m, x_n)$. and the same applies to $D_{\bar{x}}$. Their closeness is measured by $\|D_x - D_y\|_2$.

Table 1: Notation table

| Notation | Meaning |
|---|---|
| $D = \{(\mathbf{X}, \mathbf{Y})\} = \{(x_i, y_i)\}_{i=1}^t$ | a dataset with $t$ data points |
| $\mathcal{X} \subseteq \mathbb{R}^d$ | ($d$-dimensional) input space |
| $\mathcal{Y} \subseteq \mathbb{R}$ | output space |
| $\mathcal{Z} \subseteq \mathbb{R}^{d'}$ | ($d'$-dimensional) latent space, $d' \ll d$ |
| $\alpha$ | acquisition function |
| $m$ | mean function of a GP |
| $k$ | kernel function of a GP |
| $\mu_\phi$ | encoder parameterized by $\phi$ |
| $\mu_\theta$ | decoder parameterized by $\theta$ |
| $f$ | a function in original space $\mathcal{X} \to \mathcal{Y}$ |
| $g$ | a function in latent space $\mathcal{Z} \to \mathcal{Y}$ |
| $f^{\text{VAE}}$ | a reconstructed function in original space $\mathcal{X} \to \mathcal{Y}$ |
| $m_z$ | mean function of the GP in latent space |
| $k_z$ | kernel function of the GP in latent space |
| $\{\mathcal{P}_n\}_{n=1}^{d-1}$ | orthogonal polynomials |
| $\bar{x}$ | a data point in reconstructed space |
| $c_t$ | HiPPO representation of data (in original space) until $t$ |
| $\bar{c}_t$ | HiPPO representation of data (in reconstructed space) until $t$ |
| $A, B$ | HiPPO coefficient matrices |
| $\rho$ | HiPPO order |
| $\mathcal{B}$ | BO Budget |
| $\nu$ | VAE update frequency in BO |

Recall the definition of $k$-th moment of a function $f(t)$ is $\int t^k f(t) \mathrm{d}t$. These moments indeed capture the geometric shape of the $f$. According to the definition of $c_x = \left[ \int_{-\infty}^t f_x(\tau)\mathcal{P}_0(\tau)\mathrm{d}\tau, \int_{-\infty}^t f_x(\tau)\mathcal{P}_1(\tau)\mathrm{d}\tau, \ldots, \int_{-\infty}^t f_x(\tau)\mathcal{P}_{\rho-1}(\tau)\mathrm{d}\tau \right]$ where $\mathcal{P}$ are orthogonal polynomials, $c$ can be seen as linear combination of moments of $f_x$ up to order $\rho$, which is the reason why it serves as a good memory/representation of $\{x\}$ in the literature. Under same polynomial basis, if $c_x \approx c_{\bar{x}}$, then $\{x\}$ and $\{\bar{x}\}$ share approximately same moments up to order $\rho$.

Next, we are going to show that kernel distances can be represented by moments. Firstly, for certain kernels (e.g., linear $k(a, b) = a^T b$), the average pair kernel distance directly relates to moments:

$$D_x = \frac{1}{N^2} \sum_{m,n} x_m^T x_n = \left( \frac{1}{N} \sum_{m=1}^N x_m \right)^T \left( \frac{1}{N} \sum_{n=1}^N x_n \right) = \|\kappa_x\|_2^2$$

and

$$D_{\bar{x}} = \frac{1}{N^2} \sum_{m,n} \bar{x}_m^T \bar{x}_n = \left( \frac{1}{N} \sum_{m=1}^N \bar{x}_m \right)^T \left( \frac{1}{N} \sum_{n=1}^N \bar{x}_n \right) = \|\kappa_{\bar{x}}\|_2^2$$

where $\kappa_x$ is the mean of $\{x_i\}$. Thus, $\|D_x - D_{\bar{x}}\|_2 = \|\|\mu_x\|_2^2 - \|\mu_{\bar{x}}\|_2^2\|_2$, which is small if $\kappa_x \approx \kappa_{\bar{x}}$ (implied by $c_x \approx c_{\bar{x}}$ for first-order HiPPO). For higher-order kernels (e.g., polynomial of degree $p$, $k(a, b) = (a^T b + c)^p$), apply the multinomial theorem and then we have $k(a, b) = \sum_{|\alpha| \leq p} h a^\alpha b^\alpha$ so $D_x$ depends on components raised to powers up to $p$ (moments up to order $p$). Hence, if HiPPO captures these moments (i.e., $\rho \geq p$), then $c_x \approx c_{\bar{x}}$ implies $D_x \approx D_{\bar{x}}$. For the kernels out of polynomial, they can be approximated by a polynomial one with finite degree $p$, and we only need to guarantee that $\rho \geq p$, then we have $c_x \approx c_{\bar{x}}$ implies $D_x \approx D_{\bar{x}}$.

To summarise, if appropriate orthogonal polynomials are selected (e.g., $\rho \geq p$), the closeness of HiPPO representations of two sequences of data would imply their closeness in kernel distance.

### A.3 DETAILS FOR THE VISUALISATION IN FIGURE 2

In Fig. 2(a), the *x_seq* is 50 points from a *sin()* function with an additional additive random noise from the standard normal distribution with coefficient 0.1, while the *y_seq* is 50 points from a *sin()* function with an additive random noise from the standard normal distribution with coefficient 0.5. In

Fig. 2(b), the *x_seq* is 50 points from a *sin()* function with an additive random noise from the standard normal distribution with coefficient 0.1, while the *y_seq* is 50 points from a *tanh()* function with an additive random noise from the standard normal distribution with coefficient 0.5. The correlation of data points in two panels at the right-hand side in each row is calculated using a polynomial kernel (with order=1), while the HiPPO is with order 5. More examples are given in Fig. 6.

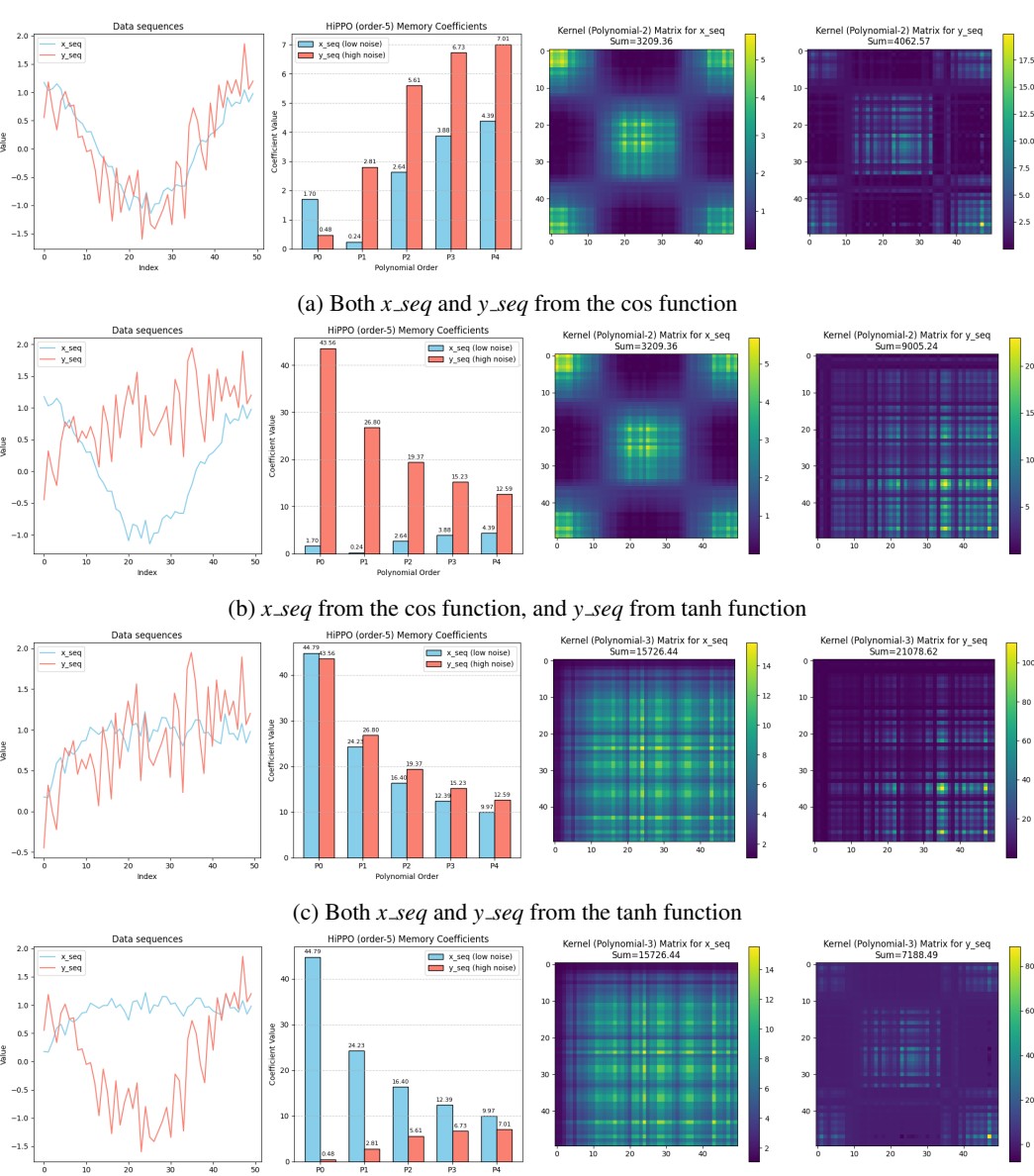

(a) Both *x_seq* and *y_seq* from the cos function

(b) *x_seq* from the cos function, and *y_seq* from tanh function

(c) Both *x_seq* and *y_seq* from the tanh function

(d) *x_seq* from the tanh function, and *y_seq* from cos function

Figure 6: Empirical demonstration of the capability of HiPPO representation in expressing the correlation between data points. The left subfigure shows two sequences of data points (i.e., *x_seq* and *y_seq*) that have roughly similar correlations due to similar functional trends (the kernel matrix between data points is also visualised in the third and fourth subfigures). As shown in the second subfigure, when the shapes or trends of two sequences are similar, their corresponding HiPPO representations are close; in contrast, their distance increases when the sequences differ.

## A.4 EXPERIMENT DETAILS

For the REWEIGH, we use the direct weighting method (instead of using weighted sampling for batches), which means the weights are directly multiplied by the loss errors of the corresponding data points. The weights are rank-based. For the GPior, we only use the target function values to evaluate the GP prior kernel values without any additional information.

Table 2: Hyperparameters or setup for Sections 5.1 and 5.2

| hyperparameter | value |
|---|---|
| VAE architecture | INPUT-FC(50)-FC(50)-FC(50)-FC(50)-FC(50)-(MU, SIGMA) |
| seed | 123 |
| the epoch number for BO | 100 |
| the initial data samples used for the first VAE pretraining | 10 |
| the epoch number VAE update | 10 |
| the acquisition function | UpperConfidenceBound |
| ode discrete method for HiPPO | 'bilinear' |
| HiPPO dimension | 50 |

Table 3: Hyperparameters or setup for Sections 5.3

| hyperparameter | value |
|---|---|
| VAE encoder | CONV(1,4)-CONV(4,8)-CONV(8,8)-CONV(8,16)-CONV(16,16) x 7-FC(256-32)-FC(32,2*latent-dim) |
| VAE decoder | FC(latent-dim,32)-FC(32, 256)-CONV(16,32)-CONV(32,32) x 2-CONV(32,16)-CONV(16,16) x 3-CONV(16-8)-CONV(8-8) x 2-CONV(8,1) |
| seed | 1 |
| query budget | 500 |
| VAE update frequency | 5 |
| ode discrete method for HiPPO | 'bilinear' |
| HiPPO dimension | 50 |

Table 4: Hyperparameters or setup for Sections 5.4

| hyperparameter | value |
|---|---|
| VAE | JTVAE |
| seed | 2 |
| query budget | 1000 |
| VAE update frequency | 50 |
| ode discrete method for HiPPO | 'bilinear' |
| HiPPO dimension | 50 |

In the molecule generation experiment, we employ the Junction Tree Variational Autoencoder (JT-VAE), which extends traditional VAEs to molecular graphs through specialized encoders and decoders. The encoder learns two distinct latent representations: one captures the tree structure and high-level cluster information, while the other encodes fine-grained connectivity details.

Molecular graph generation proceeds in two stages. First, the model constructs a tree-structured scaffold composed of chemical substructures. Then, it assembles these substructures into a complete molecule. The molecule is represented by a latent vector z=[z_T, z_G], where z_T encodes the tree structure and cluster-level information, and z_G captures the detailed connectivity between clusters. The latent space has a total dimensionality of 56, with 28 dimensions each for z_T and z_G.

Decoding also occurs in two phases. The tree decoder reconstructs the junction tree, followed by the graph decoder, which predicts the fine-grained connections between clusters to form the full molecular graph. This component-wise generation approach avoids atom-by-atom assembly and prevents chemically invalid intermediates. Given the complexity of molecular structures and their limited computability, we utilize intermediate representations from JT-VAE: x_tree_vecs and x_moles_vecs, each with 450 dimensions. Together, they form the original data space with a total dimensionality of 900.

