# OpenReview forum: "HiBBO: HiPPO-based Space Consistency for High-dimensional Bayesian Optimisation"
_ICLR.cc/2026/Conference — Submitted to ICLR 2026_

### Official Review · Reviewer_ccGD · 2025-10-20

**Soundness:** 3
**Presentation:** 3
**Contribution:** 2
**Rating:** 2
**Confidence:** 4

**Summary:**

This paper proposes HiBBO (HiPPO-based Bayesian Optimization), a method to improve high-dimensional Bayesian optimization (BO) using variational autoencoders (VAEs). The authors identify that standard VAE-based BO approaches often suffer from a distribution mismatch between the latent and original spaces: standard reconstruction losses often fail to maintain the geometric or kernel relationships between data points in the original space.
To address this, HiBBO introduces a HiPPO-based space-consistency constraint that uses High-order Polynomial Projection Operators to better align the functional distributions between the learned latent space and original space, leading to improved optimization performance in the latent space.
The paper combines this constraint with a standard BO loop and evaluates the method on benchmark tasks such as Ackley, MNIST, shape optimization, and molecular design (logP). Across all tasks, HiBBO shows faster convergence and higher objective values than prior VAE-based BO baselines (BASE, METRIC, GPior, and REWEIGH). The authors conclude that HiBBO improves the fidelity of latent representations and enhances BO efficiency across diverse high-dimensional problems.

**Strengths:**

Originality:
The paper introduces a novel idea of using HiPPO-based space consistency to preserve geometric relationships between the original and latent spaces in variational autoencoders for Bayesian optimization. This paper takes a unique approach by targeting the representation mismatch problem through a theoretically grounded regularization mechanism. The use of HiPPO, originally developed for sequence modeling, in this context is creative and represents a new application of that technique.

Quality:
The paper is technically sound, with clear motivation for the mismatch issue and a reasonable formulation of the HiPPO-based constraint. The experimental results are consistent and show measurable improvement over several established VAE-based BO baselines across multiple benchmarks.


Clarity:
The paper is clearly written and well-organized. The motivation is easy to follow, and the authors provide good intuition for how the HiPPO regularizer enforces latent–original alignment. Figures effectively illustrate both the conceptual framework and empirical results.


Significance:
The proposed idea addresses a fundamental issue that affects a broad class of latent-space optimization methods, namely, the reliability of the learned latent representation. By improving the faithfulness of this representation, the method has the potential to enhance the robustness and performance of downstream optimization across diverse high-dimensional domains, such as molecular design, robotics, and materials discovery.

**Weaknesses:**

Lack of Comparison to Recent State-of-the-Art Latent-Space BO Methods:


While the paper compares to several earlier VAE-based baselines (e.g., Gómez-Bombarelli et al., 2018; Grosnit et al., 2021; Ramchandran et al., 2025), it omits direct comparison to more recent and higher-performing latent-space Bayesian optimization (LS-BO) methods. Examples include LOL-BO (Local Latent Space Bayesian Optimization over Structured Inputs, NeurIPS 2022) and NF-BO (Latent Bayesian Optimization via Autoregressive Normalizing Flows, ICLR 2025). LOL-BO uses the same JTVAE model and LogP molecular-design benchmark as this paper, achieving substantially higher scores (> 100 vs. ~35 here, see figure 1 in LOL-BO paper). NF-BO further improves upon LOL-BO by employing a normalizing-flow rather than a VAE model. As far as I am aware, NF-BO is currently regarded as the state-of-the-art LS-BO method for molecular design tasks. Even though HiBBO introduces a novel HiPPO-based regularization mechanism within the VAE framework, a complete evaluation should include direct empirical comparisons to, and discussion of, these recent LS-BO approaches. At present, the omission of these direct comparisons makes it difficult to assess how much progress HiBBO represents relative to the existing LS-BO methods.

**Questions:**

LS-BO Baselines: Maybe I'm missing something here, is there a reason why state-of-the-art latent space Bayesian optimization (LS-BO) methods like those I mentioned above were not discussed or compared to empirically?


Reproducibility: If accepted, will the authors release the code to run their method and reproduce provided empirical results?


Computational Cost: What is the additional computational overhead introduced by the HiPPO-based term during VAE training?

---

### Official Review · Reviewer_31fY · 2025-10-27

**Soundness:** 1
**Presentation:** 1
**Contribution:** 3
**Rating:** 2
**Confidence:** 4

**Summary:**

This paper examines and proposes a solution to the "misalignment problem" in latent-space optimization$^*$ (LSO). This problem occurs when the encoder and decoder of the Variational Autoencoder (VAE) used in LSO are not properly aligned, leading to a mismatch between a surrogate function applied to the latent space (via the encoder) and the corresponding function in the input space (induced by the decoder). (This phenomenon is illustrated in Figure 1 of the paper.)

In investigating this issue, the authors mostly focus on Gaussian process–based surrogate models and show that the function can be mismatched in terms of both the means and the kernel functions; they then argue that the general reconstruction loss in a VAE works well for the means, but does not satisfactorily focus on the discrepancy between the kernel functions. The solution the authors therefore propose is to add a term to the ordinary VAE loss function. This term penalizes the distance between HiPPO (Gu et al., 2020; 2023) representations between the initial and reconstructed inputs. The authors show that this will reduce the difference between the average kernel pair distances and empirically demonstrate the advantage of their approach on some basic LSO problems.

$^*$ Latent-space optimization involves training an autoencoder to enable lower dimensional continuous optimization of a higher-dimensional and/or discrete original problem space.

**Strengths:**

### S1 Tackles an interesting problem — how to better shape the latent space in LSO problems
The problem the paper tackles (how to better shape the latent spaces used in LSO) is an interesting and actively researched problem (see, e.g., the works the paper cites in Section 4 and the additional references I have listed in this review below). Working out how to do this successfully could have beneficial downstream effects on lots of interesting optimization problems in materials and drug discovery (among others).

### S2 Compared to previous approaches the method does not require oracle labels in shaping the VAE’s latent space
Compared to previous approaches (such as Grosnit et al (2021), Tripp et al. (2020), and the methods cited below), the proposed method does not require any labels to shape the latent space. In practical problems, these labels can be expensive to require and so developing an approach that avoids this requirement seems advantageous.

> Deshwal, Aryan, and Jana Doppa. "Combining latent space and structured kernels for Bayesian optimization over combinatorial spaces." Advances in neural information processing systems 34 (2021): 8185-8200.

> Maus, Natalie, et al. "Local latent space bayesian optimization over structured inputs." Advances in neural information processing systems 35 (2022): 34505-34518.

> Eissman, Stephan, et al. "Bayesian optimization and attribute adjustment." Proc. 34th Conference on Uncertainty in Artificial Intelligence. 2018.

> Lee, Seunghun, et al. "Advancing bayesian optimization via learning correlated latent space." Advances in Neural Information Processing Systems 36 (2023): 48906-48917.

That being said, it would be nice to better understand, how the method compares to other methods that also do not require extra oracle calls such as:

> Chu, Jaewon, et al. "Inversion-based latent bayesian optimization." Advances in Neural Information Processing Systems 37 (2024): 68258-68286.

**Weaknesses:**

### W1 I found parts of the paper hard to understand
I found parts of the paper hard to understand. For instance, it would have been nice to have had at least a short introduction on how HiPPO works (e.g., in the appendix), and/or seen the differences of HiPPO-LegS vs other approaches (line 296) more thoroughly discussed. Likewise, the details of the kernels/Gaussian processes used could have been included in the main paper or appendix, to save having to search through the code for these details.

On a related note, I found some of the statements made in the paper somewhat misleading/vague. Specific examples include:
- Line 355 talks about Gómez-Bombarelli et al. (2018) introducing reweighting for the VAE. This seems more like Tripp et al. (2020)?
- Line 318 talks about Tripp et al. (2020) introducing constrained BayesOpt, whereas they introduce the reweighting/finetuning approach?
- Line 250 talks about how the closeness of HiPPO representations implies closeness of kernel distance. However, the proof in the appendix only proves this for average pair kernel distances ($D_x$). It seems it would be possible to have similar averages with very different pairs of distances?
- Line 126 talks about the variance of the kernel function equals 0 and the dimensionality tends to 0, but I did not know what this variance is over?

See also Qs 1-3 below.

### W2 Trends in the baselines do not match previous results
The trends between the baselines do not match previous results (for instance, the “Reweigh” method of Tripp et al., 2020, seems to underperform the original VAE method without any reweighted finetuning). Given that the baseline methods were run using previous work’s codebases (see line 451) this seems quite odd and, moreover, does not seem commented on in the current manuscript.

### W3 Benchmark problems are quite simple
Despite the fact that several different benchmarks are considered, these are all fairly simple. The hardest is probably logP for molecule optimization and this is not a particularly difficult or interesting problem anymore — see references below for more interesting optimization benchmarks. It would have been nice to have seen some evaluation on some harder problems to see if the method scaled.

I also did not understand the motivation behind including the shape optimization task (Section 5.3) in the experiments. This does not require training or using a GP so it seems the proposed method is doing more than just ensuring that the average kernel distances (which are not important for this case?) are consistent?

> Huang, Kexin, et al. "Therapeutics data commons: Machine learning datasets and tasks for drug discovery and development." arXiv preprint arXiv:2102.09548 (2021).

> Gao, Wenhao, et al. "Sample efficiency matters: a benchmark for practical molecular optimization." Advances in neural information processing systems 35 (2022): 21342-21357.

> Cieplinski, Tobiasz, et al. "Generative models should at least be able to design molecules that dock well: A new benchmark." Journal of Chemical Information and Modeling 63.11 (2023): 3238-3247.

> Brown, Nathan, et al. "GuacaMol: benchmarking models for de novo molecular design." Journal of chemical information and modeling 59.3 (2019): 1096-1108.

### Summary of my review.
I have gone with a lower overall score at the moment, as I believe these listed weakness (around clarity and soundness of experiments), outweigh the paper's advantages (important problem and proposed method can sidestep the requirement of needing labels for shaping latent space). Hopefully, these issues (as well as the questions below) can be addressed in the rebuttal.

**Questions:**

1. How does the backpropagation of the new loss in equation 4 actually happen? Presumably this updates the encoder/decoder weights but how does this work for discrete optimization problems such as molecules? Do you then only differentiate through part of the networks?

2. I did not follow the argument on line 194 that the general reconstruction loss in a VAE targets $\Delta^\text{mean}$ well, but not $\Delta^\text{kernel}$. I was uncertain if this is a universally applicable statement or dependent on the specific functional forms of the mean and kernel functions? In the limit, when the reconstruction term is zero (i.e., the autoencoder is perfect), both the mean and kernel function distances should be zero, which seems to contradict the presented argument?

3. One can think of the GP over the latents as actually a GP over the original inputs with a deep kernel (Wilson et al. 2016), composed of the VAE's encoder network with the RBF or other kernel used. This seems very different to the simple polynomial kernels considered in the assumptions to the proof of proposition 1, and so I wonder how much of this analysis actually holds in these more realistic situations?

> Wilson, A. G., Hu, Z., Salakhutdinov, R., & Xing, E. P. (2016, May). Deep kernel learning. In Artificial intelligence and statistics (pp. 370-378). PMLR.


4. A simple straightforward way to encode the constraint that the pairwise kernel values should be similar for encoded and decoded inputs (if that's what we want to do) would be just to add this term directly to the ordinary VAE loss (i.e., $\sum_{n,m} \| k(x_n, x_m) - k(\hat{x}_n, \hat{x}_m) \|$). Have you tried such a comparison?
 )


5. Is there a way to quantify how well the approach works (in terms of reducing kernel distances) on the harder optimization problems considered? (Rather than just looking at optimization performance). Does it come at the expense of the reconstruction loss for the VAE being worse? (I thought Figure 2 was a nice experiment/visualization in this regard on a toy problem).

---

### Official Review · Reviewer_4PKg · 2025-11-01

**Soundness:** 1
**Presentation:** 2
**Contribution:** 1
**Rating:** 2
**Confidence:** 4

**Summary:**

Traditional Variational Autoencoder-based Bayesian Optimization (VAE-BO) methods are widely used to address high-dimensional optimization problems by learning compact latent representations. This paper tackles the distribution mismatch problem, where the kernel distance in the latent space differs from that in the original input space. To resolve this issue, the authors propose HiBBO (HiPPO-based Bayesian Optimization) — a novel VAE-BO framework that integrates the HiPPO algorithm into the Bayesian Optimization process to enforce distributional consistency between the two spaces.

**Strengths:**

First attempt to incorporate HiPPO into VAE-based Bayesian Optimization.

**Weaknesses:**

**Major Weaknesses**

1. **Lack of Recent SOTA Baseline Comparisons**

    The paper does not include comparisons with several strong baselines, such as [1,2,3,4,5], which are representative recent approaches in VAE-Bayesian optimization.

2. **Lack of a Formal Definition of “Distribution Mismatch”**

    The authors seem to attribute the problem to the inability of the VAE’s reconstruction process to preserve distances between points in the latent and original spaces, but does not provide a formal definition.

3. **Lack of Formal Grounding for Kernel Discrepancy**

    The argument that the latent space distances fail to reflect the true distances in the original space is not fully convincing. As the authors themselves acknowledge, the kernel $k_x$ is not explicitly defined in the original space, so it is unclear how one can claim a measurable difference between $k_x$ and $k_z$.

4. **Weak Motivation**

    Gaussian Processes (GPs) generally assume smoothness of the function being emulated. Many prior studies thus aim to enhance smoothness between $\mathcal{Z}$ and $\mathcal{Y}$.[2,3,6] The present work appears to focus on improving smoothness between $\mathcal{Z}$ and $\mathcal{X}$, but the motivation and justification for doing so are not clearly explained.

5. **Limitations of Figure 1 to Illustrate the Problem**

    It seems that Figure 1 does not clearly illustrate the problem addressed in this paper. Figure 1 appears to highlight the reconstruction error of the VAE, which has already been addressed in [4] and [7]. However, the paper does not sufficiently consider how the issues arising from reconstruction error in VAE-BO differ from those tackled by the two previous works.


---

**Minor Weaknesses**

- **Ambiguity in Line 153:** The paper claims that the latent representations are “smooth” (Line 153). It is unclear what “smooth” precisely means in this context—whether it refers to continuity in the latent manifold, reconstruction regularity, or kernel smoothness.
- **Baseline Clarification:** The baseline named *REWEIGHT* is described as “reweighting data points to prioritize those with higher performance values.” This appears conceptually identical to *W-LBO*; the distinction should be clarified.
- **Notation:** The paper uses $\mu_\phi$ and $\mu_\theta$ for the encoder and decoder, respectively. However, it would be more standard to denote them as $q_\phi$ and $p_\theta$.
- **Figure 3:** The notation should be corrected from “$\bar x_1 \rightarrow \bar x_1$” to “$\bar x_1 \rightarrow \bar x_2$”
- **Figure 2:** The figure’s explanation is deferred to the appendix, raising the question of why it is presented in the main body.
- **Ackley Results:** The figures lack standard errors, which are essential for demonstrating statistical significance.


**Reference**

[1] Tripp, Austin, Erik Daxberger, and José Miguel Hernández-Lobato. "Sample-efficient optimization in the latent space of deep generative models via weighted retraining." Advances in Neural Information Processing Systems 33 (2020): 11259-11272.

[2] Maus, Natalie, et al. "Local latent space bayesian optimization over structured inputs." Advances in neural information processing systems 35 (2022): 34505-34518.

[3] Lee, Seunghun, et al. "Advancing bayesian optimization via learning correlated latent space." Advances in Neural Information Processing Systems 36 (2023): 48906-48917.

[4] Chu, Jaewon, et al. "Inversion-based latent bayesian optimization." Advances in Neural Information Processing Systems 37 (2024): 68258-68286.

[5] Moss, Henry B., Sebastian W. Ober, and Tom Diethe. "Return of the latent space COWBOYS: Re-thinking the use of VAEs for Bayesian optimisation of structured spaces." arXiv preprint arXiv:2507.03910 (2025).

[6] Grosnit, Antoine, et al. "High-dimensional Bayesian optimisation with variational autoencoders and deep metric learning." arXiv preprint arXiv:2106.03609 (2021).

[7] Lee, Seunghun, et al. "Latent bayesian optimization via autoregressive normalizing flows." arXiv preprint arXiv:2504.14889 (2025).

**Questions:**

See Weakness section.

---

### Official Review · Reviewer_NYNe · 2025-11-05

**Soundness:** 3
**Presentation:** 3
**Contribution:** 3
**Rating:** 4
**Confidence:** 4

**Summary:**

The main idea presented in this paper is to learn a better low dimensional representation for high dimensional bayesian optimization (HBO). HBO is often tackled by learning a lower dimensional representation for a high dimensional space and fitting a GP in the low dimensional space. The paper claims that this often leads to a significant divergence in the smoothness of the learned GP compared to the original space.

The proposed idea is to project the high-dimensional inputs into an orthogonal basis and add a regularization term on the coefficients of the orthogonal representation between the original and the reconstructed output. It is claimed that the correlations structure is closer to the original with this regularization.

Experimental results on benchmark datasets show improved performance compared to other standard HBO baselines.

**Strengths:**

- The proposed method seems to be a significant improvement compared to prior baselines on several benchmark datasets.
- The method is simple, yet practically effective showing strong improvements.

**Weaknesses:**

- The main idea is to add a regularization term to the VAE loss to ensure that the original and the reconstructed inputs, projected to an orthogonal basis are close. This is essentially projecting the difference between the original and reconstructed inputs onto the orthogonal basis. The orthogonal basis retains low frequency components and removes high frequency components - removing random noise. Thus the regularizer minimized the denoised difference.
  - It is unclear why this helps, and how it is better compared to minimizing the reconstruction loss directly in the original space without projecting it. Both losses seem to achieve a similar goal - minimizing a reconstruction error. What advantage does the projection step have?

**Questions:**

- What is the order of the orthogonal basis? how was it selected?
- Proposition 1 says that if the projections are similar then the kernels will be similar. Even if the original and reconstructed vectors are similar, their kernels are also going to be similar. How does proposition 1 demonstrate that minimizing the projection coefficient difference is better than minimizing the reconstruction loss?
- Figure 2: Are the correlation plots for the GP kernel fitted on the data? The y_seq does not change between the figures, yet the kernel correlation for y_seq (4th column) is different between figures a and b.

---

### Meta-Review · Area_Chair_acr4 · 2026-01-06

**Summary:**

This paper proposes a VAE-based high-dimensional Bayesian optimization framework that adds a space consistency regularizer to reduce latent vs original space mismatch (framed largely as a kernel-geometry mismatch).

The consensus was rejection primarily due to incomplete positioning vs recent LS-BO/ latent BO SOTA and weak experimental results (e.g., missing statistical significance justification, unconvincing results and baselines that do not align with prior reports).

**Reviewer Concerns:**

I agree with reviewers concerns in general. No rebuttal was provided and most remain outstanding.

**Reviewer Scores:**

n/a. There was no rebuttal.

---

### Decision · Program_Chairs · 2026-01-26

Reject